# Supersymmetric Polynomials on the Space of Absolutely Convergent Series

**Farah Jawad** and **Andriy Zagorodnyuk \*** 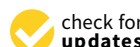

Department of Mathematical and Functional Analysis, Vasyl Stefanyk Precarpathian National University, 57 Shevchenka Str., 76018 Ivano-Frankivsk, Ukraine
**\*** Correspondence: azagorodn@gmail.com

**Abstract:** We consider an algebra $H_b^{sup}$ of analytic functions on the Banach space of two-sided absolutely summing sequences which is generated by so-called supersymmetric polynomials. Our purpose is to investigate $H_b^{sup}$ and its spectrum with using methods of infinite dimensional complex analysis and the theory of Fréchet algebras. Some algebraic bases of $H_b^{sup}$ are described. Also, we show that the spectrum of the algebra of supersymmetric analytic functions of bounded type contains a metric ring $\mathcal{M}$. We prove that $\mathcal{M}$ is a complete metric (nonlinear) space and investigate homomorphisms and additive operators on this ring. Some possible applications are discussed.

**Keywords:** symmetric and supersymmetric polynomials on Banach spaces; algebras of analytic functions on Banach spaces; spectra algebras of analytic functions

## 1. Introduction and Preliminaries

Let $X$ be a complex Banach space. A (continuous) map $P\colon X \to \mathbb{C}$ is said to be a (continuous) $n$-homogeneous polynomial if there exists a (continuous) $n$-linear map $B_P\colon X^n \to \mathbb{C}$ such that $P(x) = B_P(x, \ldots, x)$. 0-homogeneous polynomial is just a constant function. A finite sum of homogeneous polynomials is a polynomial. We denote by $\mathcal{P}(^nX)$ the space of all continuous $n$-homogeneous polynomials on $X$ and by $\mathcal{P}(X)$ the space of all polynomials on $X$. Note that $\mathcal{P}(^nX)$ is a Banach space with respect to any of the norms

$$\|P\|_r = \sup_{\|x\| \le r} |P(x)|, \qquad r > 0. \tag{1}$$

Let $\tau_b$ be the topology on $\mathcal{P}(X)$ of uniform convergence on bounded subsets of $X$. This topology is generated by the countable family of norms (1) for positive rational numbers $r$ and so is metrisable. We denote by $H_b(X)$ the completion of $(\mathcal{P}(X), \tau_b)$. So $H_b(X)$ is a Fréchet algebra which consists of entire analytic functions on $X$ which are bounded on all bounded subsets (so-called *entire functions of bounded type*). For details on polynomials and analytic functions on Banach spaces we refer the reader to [1]. The spectra (sets of continuous complex homomorphisms = sets of characters) of $H_b(X)$ and its subalgebras were investigated by many authors (see e.g., [2–5]).

Let $G$ be a group of isometric operators on $X$. We denote by $H_{bG}(X)$ the subalgebra of $H_b(X)$ which consists of $G$-invariant analytic functions. Such algebras were considered in the general case in [6,7]. For some special cases of $G$ there is a sequence of $G$-symmetric homogeneous polynomials

$\{P_1, P_2, \ldots, P_n, \ldots\}$, $\deg P_n = n$ which forms an algebraic basis in the algebra of $G$-symmetric polynomials $\mathcal{P}_G(X)$. For example, if $G = s$ is the group of all permutations of the basis vectors in $\ell_1$, then the functions

$$F_k(x) = \sum_{i=1}^{\infty} x_i^k, \qquad k \in \mathbb{N}$$

form an algebraic basis in $\mathcal{P}_s(\ell_1)$ [8]. The following bases in $\mathcal{P}_s(\ell_1)$ also are important

$$G_n(x) = \sum_{i_1 < \cdots < i_n} x_{i_1} \cdots x_{i_n}$$

and

$$H_n(x) = \sum_{i_1 \leq \cdots \leq i_n} x_{i_1} \cdots x_{i_n}.$$

Let $\mathcal{F}(x)(t)$, $\mathcal{G}(x)(t)$ and $\mathcal{H}(x)(t)$ be formal series

$$\mathcal{F}(x)(t) = \sum_{n=1}^{\infty} t^{n-1} F_n(x),$$

$$\mathcal{G}(x)(t) = \sum_{n=0}^{\infty} t^n G_n(x), \qquad G_0 = 1$$

and

$$\mathcal{H}(x)(t) = \sum_{n=0}^{\infty} t^n H_n(x), \qquad H_0 = 1$$

which also are called generating functions. From combinatorial considerations it is known ([9] p. 3) that

$$\mathcal{G}(x)(t) = \frac{1}{H(-x)(t)} \tag{2}$$

and

$$\mathcal{G}(x)(t) = \exp\left(-\sum_{n=1}^{\infty} t^n \frac{F_n(-x)}{n}\right) = \exp\left(-\int_0^t \mathcal{F}(-x)(\xi)d\xi\right), \tag{3}$$

where the equality holds for every $x \in \ell_1$ and every $t$ in the common domain of convergence. In [10] it is shown that every complex homomorphism $\varphi$ of $H_{bs}(\ell_1)$ is completely defined by its value on $\mathcal{G}(x)(t)$ and

$$g(t) = \varphi(\mathcal{G}(t)) = \sum_{n=0}^{\infty} t^n \varphi(G_n) \tag{4}$$

is a function of exponential type with $g(0) = 1$. Moreover, if $\varphi = \delta_x$ is the point evaluation functional at $x \in \ell_1$ (that is $\delta_x(f) = f(x)$, $f \in H_b(\ell_1)$), then

$$\delta_x(\mathcal{G}(t)) = \mathcal{G}(x)(t) = \prod_{k=1}^{\infty}(1 + x_k t). \tag{5}$$

Note that (5) is an absolutely convergent Hadamard Product—the entire function defined by its zeros $a_n = 1/(-x_n)$ for $x_n \neq 0$. Also [10,11], there is a family $\psi_\lambda$, $\lambda \in \mathbb{C}$ in the spectrum of $H_b(\ell_1)$ such that

$$\psi_\lambda(\mathcal{G}(t)) = e^{\lambda t}.$$

In [12] it is shown that there is a function of exponential type $\gamma$ with $\gamma(0) = 1$ but which cannot be represented as in (4). Spectra of algebras $H_{bs}(\ell_p)$ were investigated also in [13,14]. Polynomials which are symmetric with respect to some other representations of the group of permutations of natural numbers were considered in [15–17].

In this paper we consider a subalgebra of entire functions of bounded type which is generated by so-called supersymmetric polynomials. Algebras of supersymmetric polynomials on finite-dimensional spaces were considered in [18–20]. In Section 2.1 we consider some important bases in the algebra of supersymmetric polynomials. Section 2.2 is devoted to the spectrum of the algebra of supersymmetric analytic functions of bounded type. In particular, we show that the set of point evaluation functionals on the algebra can be described as a metric ring which is not a linear space. Some operators on this ring are investigated.

## 2. Results

### 2.1. Bases of Supersymmetric Polynomials

We will use $\mathbb{N}$ for natural numbers and $\mathbb{Z}$ for integers. Also, we set $\mathbb{Z}_0 = \mathbb{Z} \setminus 0$ and denote by $\ell_1(\mathbb{Z}_0)$ the Banach space of all absolutely summing complex sequences indexed by numbers in $\mathbb{Z}_0$. The symbol $\ell_1 = \ell_1(\mathbb{N})$ means the classical Banach space of absolutely summing complex sequences. Any element $z$ in $\ell_1(\mathbb{Z}_0)$ has the representation

$$z = (\ldots, z_{-n}, \ldots, z_{-2}, z_{-1}, z_1, z_2, \ldots, z_n, \ldots)$$

$$= (y|x) = (\ldots, y_n, \ldots, y_2, y_1 | x_1, x_2, \ldots, x_n, \ldots)$$

with

$$\|z\| = \sum_{i=-\infty}^{\infty} |z_i|,$$

where $x = (x_1, x_2, \ldots, x_n, \ldots)$ and $y = (y_1, y_2, \ldots, y_n, \ldots)$ are in $\ell_1$, $z_n = x_n$, $z_{-n} = y_n$ for $n \in \mathbb{N}$ and

$$x \longmapsto (0|x_1, x_2, \ldots, x_n, \ldots) \text{ and } y \longmapsto (\ldots, y_{-n}, \ldots, y_{-2}, y_{-1}|0)$$

are natural isometric embeddings of the copies of $\ell_1$ into $\ell_1(\mathbb{Z}_0)$.

Let us define the following polynomials on $\ell_1(\mathbb{Z}_0)$:

$$T_k(z) = F_k(x) - F_k(y) = \sum_{i=1}^{\infty} x_i^k - \sum_{i=1}^{\infty} y_i^k, \quad k \in \mathbb{N}.$$

**Definition 1.** *A polynomial P on $\ell_1(\mathbb{Z}_0)$ is said to be supersymmetric if it can be represented as an algebraic combination of polynomials $\{T_k\}_{k=1}^{\infty}$. In other words, P is a finite sum of finite products of polynomials in $\{T_k\}_{k=1}^{\infty}$ and constants. We denote by $\mathcal{P}_{sup}$ the algebra of all supersymmetric polynomials on $\ell_1(\mathbb{Z}_0)$.*

Note first that polynomials $T_k$ are algebraically independent because $F_k$ are so. Hence $\{T_k\}_{k=1}^{\infty}$ forms an algebraic basis in $\mathcal{P}_{sup}$.

We say that $z \sim w$, for some $z, w \in \ell_1(\mathbb{Z}_0)$ if $T_k(z) = T_k(w)$ for every $k \in \mathbb{N}$. Let us denote by $\mathcal{M}$ the quotient set $\ell_1(\mathbb{Z}_0)/ \sim$ which is a natural domain for supersymmetric polynomials. For a given $z \in \ell_1(\mathbb{Z}_0)$, let $[z] \in \mathcal{M}$ be the class of equivalence which contains $z$.

Similarly like in [10] we introduce an operation "$\bullet$" on $\ell_1(\mathbb{Z}_0)$:

$$z \bullet w = (y \bullet v | x \bullet u) = (\ldots, v_n, y_n, \ldots, v_1, y_1 | x_1, u_1, \ldots, x_n, u_n, \ldots),$$

where $z = (y|x)$ and $w = (v|u)$. Also, we denote $z^- = (y|x)^- = (x|y)$. Clearly, $(z^-)^- = z$ and $z \bullet z^- \sim (0|0)$. These operations can be naturally defined on $\mathcal{M}$ by

$$[z] \bullet [w] = [z \bullet w] \text{ and } [z]^- = [z^-]. \tag{6}$$

**Theorem 1.** *The following statements hold:*

1. $T_k(z \bullet w) = T_k(z) + T_k(w)$ *for every* $k \in \mathbb{N}$.
2. *The operations in* (6) *are well defined, that is, they do not depend on the choice of representatives.*
3. $(\mathcal{M}, \bullet, [z] \mapsto [z]^-)$ *is a commutative group with zero* $0 = (0|0)$.
4. $z \sim 0$ *if and only if there are* $d, s \in \ell_1$ *such that* $z = (d|s)$ *and* $F_k(d) = F_k(s)$ *for all* $k \in \mathbb{N}$. *Equivalently, all nonzero coordinates of* $d$ *coincides with nonzero coordinates of* $s$ *up to a permutation.*

**Proof.** Assertions (1)–(3) are straightforward consequences of definitions. In [13] is proved that for given $d, s \in \ell_1$, $F_k(d) = F_k(s)$ for all $k \in \mathbb{N}$ if and only if all nonzero coordinates of $d$ coincides with nonzero coordinates of $s$ up to a permutation. $\square$

Let $\mathcal{P}_1$ and $\mathcal{P}_2$ be some algebras of polynomials on linear spaces $X$ and $Y$ respectively such that $\mathcal{P}_1$ is generated by an algebraic basis $\{P_1, P_2, \ldots, P_n, \ldots\}$ and $\mathcal{P}_2$ is generated by an algebraic basis $\{Q_1, Q_2, \ldots, Q_n, \ldots\}$ with $\deg P_n = \deg Q_n = n$, $n \in \mathbb{N}$. Then the map, defined on the basic vectors by $P_n \longmapsto Q_n$ and extended to $\mathcal{P}_1$ by linearity and multiplicativity, obviously is an algebraic isomorphism from $\mathcal{P}_1$ onto $\mathcal{P}_2$ which preserves degrees of polynomials.

Let us denote by $\Lambda$ the isomorphism from $\mathcal{P}_s = \mathcal{P}_s(\ell_1)$ to $\mathcal{P}_{sup}$ such that

$$\Lambda \colon F_n \longmapsto T_n, \qquad n \in \mathbb{N}.$$

**Proposition 1.** *If* $\{P_n\}_{n=1}^{\infty}$ *is an algebraic basis in* $\mathcal{P}_s$, *then* $\{\Lambda(P_n)\}_{n=1}^{\infty}$ *is an algebraic basis in* $\mathcal{P}_{sup}$.

**Proof.** The proof follows from the general fact that the range of any algebraic basis under an isomorphism is an algebraic basis. Indeed, $\Lambda(P_n)$, $n \in \mathbb{N}$ are algebraically independent because $P_n$, $n \in \mathbb{N}$ are so and $\Lambda$ is injective. Also, every $Q \in \mathcal{P}_{sup}$ belongs to the algebraic combination of $\{\Lambda(P_n)\}_{n=1}^{\infty}$ because $\Lambda^{-1}(Q)$ belongs to the algebraic combination of $\{P_n\}_{n=1}^{\infty}$ and $\Lambda$ is surjective (cf. [13]). $\square$

Let $H_b^{sup}$ be the completion of $\mathcal{P}_{sup}$ with respect to the topology of uniform convergence on bounded subsets. In other words, $H_b^{sup}$ is the minimal closed subspace of $H_b(\ell_1(\mathbb{Z}_0))$ which contains $\mathcal{P}_{sup}$. Elements of $H_b^{sup}$ will be called *supersymmetric analytic* or *entire* functions on $\ell_1(\mathbb{Z}_0)$.

**Proposition 2.** *The map* $\Lambda^{-1}$ *is continuous and can be extended to a continuous homomorphism from* $H_b^{sup}$ *to* $H_{bs} = H_{bs}(\ell_1)$ *with a dense range. The map* $\Lambda$ *is discontinuous and densely defined on* $H_{bs}$.

**Proof.** Let us observe first that $\Lambda^{-1}(P)$ is the restriction of $P \in \mathcal{P}_{sup}$ onto the closed subspace $\{(0|x) \colon x \in \ell_1\}$. The operator of the restriction is obviously continuous on $H_b^{sup}$ and is the extension of $\Lambda^{-1}$. The range of $\Lambda^{-1}$ is dense because it contains all symmetric polynomials on $\ell_1$. On the other hand, in [10] it is proved that the homomorphism $\Lambda_- \colon \mathcal{P}_s \to \mathcal{P}_s$ such that $\Lambda_- F_k = -F_k$, $k \in \mathbb{N}$ is discontinuous on $(\mathcal{P}_s, \tau_b)$. Moreover, in [21] a function $g(x) \in H_{bs}$ such that $\Lambda_-(g) \notin H_{bs}$ was constructed. If $\Lambda$ is continuous, it can be extended to the whole space $H_{bs}$ and so $\Lambda g(x|0) = (\Lambda_- g)(x)$. It leads to

a contradiction because on the left side we have a bounded function on all bounded subsets but on the right side, it is not so. □

For a given $y \in \ell_1$ we denote by $\Lambda_y P(x) = (\Lambda P)(y|x)$, $P \in \mathcal{P}_s$. It is easy to see that

$$\Lambda_y P(x \bullet y) = (\Lambda P)(y|x \bullet y) = (\Lambda P)(0|x) = P(x).$$

**Theorem 2.** *Let* $\Lambda G_n = W_n$. *Then*

$$W_n(y|x) = \sum_{k=0}^{n} G_k(x) H_{n-k}(-y), \qquad n \in \mathbb{N} \tag{7}$$

*and*

$$\mathcal{W}(y|x)(t) = \sum_{n=0}^{\infty} t^n W_n(y|x) = \frac{\mathcal{G}(x)(t)}{\mathcal{G}(y)(t)}, \tag{8}$$

*where the equality is true on the common domains of convergence.*

**Proof.** In [10] it is proved that

$$\mathcal{G}(x \bullet y)(t) = \mathcal{G}(x)(t)\mathcal{G}(y)(t), \qquad x, y \in \ell_1. \tag{9}$$

Hence, for a fixed $y \in \ell_1$

$$\Lambda_y\big(\mathcal{G}(x \bullet y)(t)\big) = \sum_{n=0}^{\infty} t^n (\Lambda G_n)(y|x) \sum_{n=0}^{\infty} t^n G_n(y) = \sum_{n=0}^{\infty} t^n W_n(y|x) \sum_{n=0}^{\infty} t^n G_n(y).$$

On the other hand,

$$\Lambda_y\big(\mathcal{G}(x \bullet y)(t)\big) = \sum_{n=0}^{\infty} t^n G_n(x).$$

So

$$\sum_{n=0}^{\infty} t^n W_n(y|x) \sum_{n=0}^{\infty} t^n G_n(y) = \sum_{n=0}^{\infty} t^n G_n(x). \tag{10}$$

From (10) we have

$$\mathcal{W}(y|x)(t)\mathcal{G}(y)(t) = \mathcal{G}(x)(t)$$

and so (8) holds. Taking into account Formula (2) we have

$$\sum_{n=0}^{\infty} t^n W_n(y|x) = \sum_{n=0}^{\infty} t^n G_n(x) \sum_{n=0}^{\infty} t^n H_n(-y) = \sum_{n=0}^{\infty} t^n \sum_{k=0}^{n} G_k(x) H_{n-k}(-y).$$

From here we have (7). □

**Corollary 1.**
$$\mathcal{W}((y|x) \bullet (d|b))(t) = \mathcal{W}(y|x)(t)\mathcal{W}(d|b)(t), \quad x, y, b, d \in \ell_1.$$

**Proof.** The required statement immediately follows if we combine Formulas (9) and (8). □

**Corollary 2.** *For every $n \in \mathbb{N}$ and $x, y, b, d \in \ell_1$ we have*

$$W_n((y|x) \bullet (d|b)) = \sum_{k=0}^{n} W_k(y|x) W_{n-k}(d|b),$$

$$G_n(x \bullet b) = \sum_{k=0}^{n} G_k(x) G_{n-k}(b)$$

*and*

$$H_n(y \bullet d) = \sum_{k=0}^{n} H_k(y) H_{n-k}(d).$$

**Proof.** From Corollary 1 we have

$$\sum_{n=0}^{\infty} t^n W_n((y|x) \bullet (d|b)) = \sum_{k=0}^{\infty} t^k W_k(y|x) \sum_{j=0}^{\infty} t^j W_j(d|b).$$

Taking coefficients of $t^n$ we have the first equality. The second and thirds equalities we can obtain by the same reasoning. $\square$

It is clear that $(y \bullet a | x \bullet a) \sim (y|x)$ for all $x, y, a \in \ell_1$. We say that $(y|x)$ is an *irreducible* representative of $u \in \mathcal{M}$ if $[(y|x)] = u$ and for every $x_n \neq 0$ and every $k$, $x_n \neq y_k$.

**Proposition 3.** $(y|x)$ *is irreducible if and only if $\mathcal{G}(x)(t)$ and $\mathcal{G}(y)(t)$ have no common zeros.*

**Proof.** According to (5), for nonzero elements $x_k$ and $y_k$ the numbers $(-x_k)^{-1}$ and $(-y_k)^{-1}$ are zeros of $\mathcal{G}(x)(t)$ and $\mathcal{G}(y)(t)$ respectively. $\square$

**Corollary 3.** *Let $u \in \mathcal{M}$. Then $u$ is completely defined by $\mathcal{W}(u)(t) = \mathcal{W}(y|x)(t)$ and $\mathcal{W}(y|x)(t)$ is a meromorphic functions of the form $f(t)/g(t)$ such that $f, g$ are entire functions of exponential type with $f(0) = 1$ and $g(0) = 1$, where $(y|x) \in u$. Moreover, let $(\alpha_k)$ and $\beta_k$ be zeros of $f$ and $g$ respectively. Then both $(1/\alpha_k)$ and $(1/\beta_k)$ belong to $\ell_1$,*

$$f(t) = \prod_{k=1}^{\infty} \left(1 - \frac{t}{\alpha_k}\right) \quad \text{and} \quad g(t) = \prod_{k=1}^{\infty} \left(1 - \frac{t}{\beta_k}\right),$$

*and*

$$\left(\ldots, -\frac{1}{\beta_n}, \ldots, -\frac{1}{\beta_2}, -\frac{1}{\beta_1} \middle| -\frac{1}{\alpha_1}, -\frac{1}{\alpha_2}, \ldots, -\frac{1}{\alpha_n}, \ldots\right)$$

*is an irreducible representation of $u$.*

Let $x \in \ell_1$. We denote by $\operatorname{supp} x$ the support of $x$, that is,

$$\operatorname{supp} x = \{n \in \mathbb{N} : x_n \neq 0\}.$$

**Corollary 4.** *Let $(y|x)$ and $(y'|x')$ be two irreducible representatives of $u$. Then there are bijections $i \colon \operatorname{supp} x \to \operatorname{supp} x'$ and $j \colon \operatorname{supp} y \to \operatorname{supp} y'$ such that $x_n = x'_{i(n)}$ and $y_m = y'_{j(m)}$ for all $n \in \operatorname{supp} x$ and $m \in \operatorname{supp} y$.*

**Proposition 4.** *For every $u = [(y|x)] \in \mathcal{M}$ the following equality holds on the common domain of convergence*

$$\mathcal{W}(u)(t) = \exp\left(-\sum_{n=1}^{\infty} t^n \frac{T_n(-u)}{n}\right) = \exp\left(-\int_0^t \mathcal{T}(-u)(\xi)d\xi\right), \tag{11}$$

*where $-u = [(-y| - x)]$.*

**Proof.** From (8) and (5) it follows that $\mathcal{W}(u)(t)$ converges for every $t \in \mathbb{C}$ if $y = 0$ and in the ball $|t| < r$, where

$$r = \min_{|y_n| \neq 0} |y_n|^{-1}$$

if $y \neq 0$. Since $\Lambda^{-1}$ is a continuous homomorphism, from (3) we have that for each $t \in \mathbb{C}$ such that $\mathcal{W}(u)(t)$ converges

$$\Lambda^{-1}\mathcal{W}(u)(t) = \mathcal{G}(x)(t) = \exp\left(-\sum_{n=1}^{\infty} t^n \frac{F_n(-x)}{n}\right).$$

Since $\|F_n\| = 1$, the series

$$\sum_{n=1}^{\infty} t^n \frac{F_n(-x)}{n}$$

converges if $|t| < \|x\|$. Also, $\|T_n\| = 1$ and the series

$$\sum_{n=1}^{\infty} t^n \frac{T_n(-u)}{n}$$

converges if $|t| < \|u\|$. So in the common domain of convergence

$$\exp\left(-\sum_{n=1}^{\infty} t^n \frac{F_n(-x)}{n}\right)$$

is in the domain of $\Lambda$ and

$$\mathcal{W}(u)(t) = \Lambda\mathcal{G}(x)(t) = \Lambda \exp\left(-\sum_{n=1}^{\infty} t^n \frac{F_n(-x)}{n}\right).$$

Also, in the domain

$$\Lambda^{-1} \exp\left(-\sum_{n=1}^{\infty} t^n \frac{T_n(-u)}{n}\right) = \exp\left(-\sum_{n=1}^{\infty} t^n \frac{F_n(-x)}{n}\right).$$

□

**Theorem 3.** *Let $u \in \mathcal{M}$ and $u \neq 0$. For a given $\lambda \in \mathbb{C}$ there is $v \in \mathcal{M}$ such that $T_k(v) = \lambda T_k(u)$ if and only if $\lambda$ is an integer number.*

**Proof.** Let $\lambda = m \in \mathbb{Z}$. If $n = 0$, then $v = 0$. If $n > 0$, then $v = \underbrace{u \bullet \cdots \bullet u}_{n}$. If $n < 0$, then $v = \underbrace{u^- \bullet \cdots \bullet u^-}_{n}$.

Let now $\lambda \notin \mathbb{Z}$. According to (11)

$$\mathcal{W}(v)(t) = (\mathcal{W}(u)(t))^\lambda.$$

But it contradicts representation (8) for $v$.   $\square$

## 2.2. The Spectrum of $H_B^{Sup}$ and the Nonlinear Normed Ring $\mathcal{M}$

### 2.2.1. The Spectrum

Let us denote by $M_b^{sup}$ the spectrum of $H_b^{sup}$, that is, the set of all continuous nonzero complex homomorphisms (characters) of $H_b^{sup}$. Clearly for every point $u \in \mathcal{M}$ there is a character $\delta_u \in M_b^{sup}$ (so-called point evaluation functional) such that $\delta_u(f) = f(u)$, $f \in H_b^{sup}$. Moreover, if $u \neq v$, then $\delta_u \neq \delta_v$. In this sense, we can say that $\mathcal{M} \subset M_b^{sup}$.

Since polynomials $\{W_n\}$ form an algebraic basis in $H_b^{sup}$, any character $\varphi \in M_b^{sup}$ is completely defined by its values on $W_n$, $n \in \mathbb{N}$. In other words, every character $\varphi$ can be represented by the function

$$\varphi(\mathcal{W}(t)) = \sum_{n=0}^{\infty} t^n \varphi(W_n). \tag{12}$$

Note that if $\varphi = \delta_u$ for some $u \in \mathcal{M}$, then $\varphi(\mathcal{W}(t))$ can be described by Corollary 3. Using ideas in [11,13] it is possible to construct a character which is not a point-evaluation functional. Let $\lambda$ and $\mu$ be complex numbers. Consider

$$u_n = \left(0, \ldots, 0, \frac{\mu}{n}, \ldots, \frac{\mu}{n} \Big| \frac{\lambda}{n}, \ldots, \frac{\lambda}{n}, 0, \ldots, 0\right).$$

From the compactness reasons, we have that $\{\delta_{u_n}\}$ has a cluster point $\psi_{\lambda,\mu}$ in $M_b^{sup}$. So

$$\psi_{\lambda,\mu}(\mathcal{W}(t)) = \lim_{n \to \infty} \frac{\sum_{k=0}^{\infty} t^k G_k(\lambda/n, \ldots, \lambda/n, 0, \ldots, 0)}{\sum_{k=0}^{\infty} t^k G_k(\mu/n, \ldots, \mu/n, 0, \ldots, 0)}.$$

Taking into account [10] that

$$\lim_{n \to \infty} \sum_{k=0}^{\infty} t^k G_k(\lambda/n, \ldots, \lambda/n, 0, \ldots, 0) = e^{\lambda t}$$

we have

$$\psi_{\lambda,\mu}(\mathcal{W}(t)) = e^{(\lambda-\mu)t}.$$

Comparing the representation with Corollary 3, we can see that $\psi_{\lambda,\mu}$ cannot be equal to a point evaluation functional.

### 2.2.2. The Normed Ring Structure of $\mathcal{M}$

We consider the set $\mathcal{M}$ more detailed. Let $\mathcal{M}_+ = \{u \in \mathcal{M} : u = [(0|x)], x \in \ell_1\}$. According to [12] we introduce an operation '$\diamond$' on $\mathcal{M}_+$ and extend it to $\mathcal{M}$.

Let $x, y \in \ell_1$. Then $x \diamond y$, we mean the resulting sequence of ordering the set $\{x_i y_j : i, j \in \mathbb{N}\}$ with one single index in some fixed order. If $u = [(0|x)]$ and $v = [(0|y)]$, then $u \diamond v = [(0|x \diamond y)]$. From [12,22] we know that the operation on $\mathcal{M}_+$ is commutative, associative and $[y \diamond (x \bullet d)] = [(y \diamond x) \bullet (y \diamond d)]$. Finally, let $u = [(y|x)]$ and $v = [(d|b)]$ are in $\mathcal{M}$. We define

$$u \diamond v = [((y \diamond b) \bullet (x \diamond d)|(y \diamond d) \bullet (x \diamond b))].$$

**Proposition 5.** *For every* $k \in \mathbb{N}$, $T_k(u \diamond v) = T_k(u)T_k(v)$, $u, v \in \mathcal{M}$.

**Proof.** From [12] we know that for all $x, z \in \ell_1$, $F_k(x \diamond z) = F_k(x) F_k(z)$. Let $u = [(y|x)]$ and $v = [(d|b)]$. Then

$$T_k(u \diamond v) = F_k((y \diamond d) \bullet (x \diamond b)) - F_k((y \diamond b) \bullet (x \diamond d))$$

$$= F_k(y) F_k(d) + F_k(x) F_k(b) - F_k(y) F_k(b) - F_k(x) F_k(d) = T_k(u) T_k(v).$$

□

**Theorem 4.** $(\mathcal{M}, \bullet, \diamond)$ *is a commutative ring with zero* $0 = [(0|0)]$ *and unity* $\mathbb{I} = [(0|1, 0, \dots)]$.

**Proof.** Note first that $(\mathcal{M}, \bullet)$ is a commutative group and if $u = [(y|x)] \in \mathcal{M}$, then $u^- = [(y|x)^-] = [(x|y)]$ is the inverse of $u$. The associativity and commutativity of the multiplication and the distributive low were proved in [12] for the case $\mathcal{M}_+$ and can be checked for the general case by simple computations. □

Note that there is an operation of multiplication by a constant on $\mathbb{C} \times \mathcal{M}$:

$$\lambda[(y|x)] = [(\lambda y | \lambda x)], \quad \lambda \in \mathbb{C}, \quad [(y|x)] \in \mathcal{M}.$$

Clearly,

$$\lambda(u \bullet v) = \lambda u \bullet \lambda v \quad \text{and} \quad \lambda(u \diamond v) = (\lambda u) \diamond v = u \diamond (\lambda v) \quad \lambda \in \mathbb{C}, \quad u, v \in \mathcal{M}.$$

But, in the general case,

$$(\lambda_1 + \lambda_2) u \neq \lambda_1 u \bullet \lambda_2 u.$$

So $(\mathcal{M}, \bullet, (\lambda, u) \mapsto \lambda u)$ is not a linear space over $\mathbb{C}$. Hence $(\mathcal{M}, \bullet, \diamond)$ is not an algebra. In order to topologise $\mathcal{M}$, we can use the standard norm on $\ell_1(\mathbb{Z}_0)$.

**Definition 2.** *Let* $u \in \mathcal{M}$. *We define a norm of* $u$ *by the following way:*

$$\|u\| = \|x\| + \|y\| = \sum_{n=1}^{\infty} |x_n| + \sum_{n=1}^{\infty} |y_n|,$$

*where* $(y|x)$ *is an irreducible representative of* $u$.

From Corollary 4 it follows that the definition of norm $\| \cdot \|$ does not depend on the irreducible representative. The next proposition shows that, like in a linear space, the norm has natural properties.

**Proposition 6.** *Let* $u, v \in \mathcal{M}$, $\lambda \in \mathbb{C}$ *The following properties hold:*

1.  $\|u\| \geq 0$ *and* $\|u\| = 0$ *if and only if* $u = 0$.
2.  $\|\lambda u\| = |\lambda| \|u\|$.
3.  $\|u \bullet v\| \leq \|u\| + \|v\|$.
4.  $\|u \diamond v\| \leq \|u\| \|v\|$.
5.  $\|u^-\| = \|u\|$.
6.  $\|u\| = \min_{(y|x) \in u} (\|x\| + \|y\|)$.

**Proof.** We need to prove just item (6). Let $(y|x)$ be a representation of $u$. We can write up to a permutation that $(y|x) = (y' \bullet a | x' \bullet a)$ for some $a \in \ell_1$ and irreducible $(y'|x')$. So

$$\|u\| = \|x'\| + \|y'\| \leq \|x\| + \|y\|$$

for ever $(y|x) \in u$.  □

We define a metric $\rho$ on $\mathcal{M}$, associated with the norm by the natural way. Let $u, v \in \mathcal{M}$. Set

$$\rho(u, v) = \|u \bullet v^-\|.$$

It is easy to check that $\rho$ is a metric using the same arguments as in the classical case of linear normed spaces.

**Proposition 7.** *The multiplication by $\lambda \in \mathbb{C}$, $\lambda \mapsto \lambda u$ for a fixed $u \in \mathcal{M}$ is discontinuous in general at each nonzero point in $\mathbb{C}$ and continuous at zero. Here we consider the standard topology on $\mathbb{C}$ and the topology on $\mathcal{M}$, generated by $\rho$.*

**Proof.** Let $\varepsilon_n$ be a sequence in $\mathbb{C}$ such that $\varepsilon_n \neq 0$, $\varepsilon_n \to 0$ as $n \to \infty$, $\lambda \neq 0$ and $u = [(\ldots, 0, y_1|x_1, 0, \ldots)]$, where $x_1 \neq 0$ or $y_1 \neq 0$ and $x_1 \neq y_1$. Then

$$\rho(\lambda(1 + \varepsilon_n)u, \lambda u) = \|[(\ldots, 0, \lambda y_1, \lambda(1 + \varepsilon_n)x_1|\lambda x_1, \lambda(1 + \varepsilon_n)y_1, 0, \ldots)]\|$$

$$= \|\lambda x_1\| + \|\lambda y_1\| + \|\lambda(1 + \varepsilon_n)x_1\| + \|\lambda(1 + \varepsilon_n)y_1\| > |\lambda|\|u\| > 0$$

while $\lambda(1 + \varepsilon_n) \to \lambda$ as $n \to \infty$.

Let now $\lambda = 0$, $u \in \mathcal{M}$ and $(y|x)$ be an irreducible representation of $u$. Then

$$\rho(\varepsilon_n u, 0) = \|\varepsilon_n u\| = |\varepsilon_n|\|x\| + |\varepsilon_n|\|y\| \to 0.$$

□

**Theorem 5.** *The operations '$\bullet$' and '$\diamond$' are jointly continuous on $(\mathcal{M}, \rho)$.*

**Proof.** It is easy to check that if $\rho(u, u') < \varepsilon_1$ and $\rho(v, v') < \varepsilon_2$, then

$$\rho(u \bullet v, u' \bullet v') < \|(u \bullet v) \bullet (u' \bullet v')^-\| < \varepsilon_1 + \varepsilon_2$$

and

$$\rho(u \diamond v, u' \diamond v') < \varepsilon_1 \|u\| + \varepsilon_2 \|v\| + \varepsilon_1 \varepsilon_2.$$

□

**Proposition 8.** *The metric space $(\mathcal{M}, \rho)$ is nonseparable.*

**Proof.** Let us consider the following set

$$S_1 = \{u_\lambda = \lambda \mathbb{I} = (0|\lambda, 0, 0 \ldots) : \lambda \in \mathbb{C}, |\lambda| = 1\}.$$

If $\lambda_1 \neq \lambda_2$, then

$$\rho(u_{\lambda_1}, u_{\lambda_2}) = \|(\ldots, 0, 0, \lambda_2|\lambda_1, 0, 0 \ldots)\| = 2.$$

So the unit sphere of $(\mathcal{M}, \rho)$ contains an uncountable set $S_1$ such that the distance between each pair of distinct points of $S_1$ is equal to 2.  □

**Theorem 6.** *The metric space $(\mathcal{M}, \rho)$ is complete.*

**Proof.** Let $u$ and $v$ be in $\mathcal{M}$ and $\rho(u,v) = \|u \bullet v^-\| < \varepsilon$ and $(y|x)$ be an irreducible representations of $u$. Then there is an irreducible representation $(d|b)$ of $v$ such that $\|(y|x) - (d|b)\| < \varepsilon$. Indeed the inequality $\|u \bullet v^-\| < \varepsilon$ implies that there is $w \in \mathcal{M}$ such that $u = u' \bullet w$, $v = v' \bullet w$ and $\|u'\| + \|v'\| < \varepsilon$. Let us consider a representation $(d|b)$ of $v$ such that the element $w$ in $(d|b)$ is represented by the same vector that in $(y|x)$. Let $(y'|x')$ be the irreducible representation of $u'$ in $(y|x)$ and $(d'|b')$ be the irreducible representation of $v'$ in $(d|b)$. Then

$$\|(y|x) - (d|b)\| = \|(y'|x') - (d'|b')\| \leq \|u'\| + \|v'\| < \varepsilon.$$

Let $u^{(m)}$, $m \in \mathbb{N}$ be a Cauchy sequence in $(\mathcal{M}, \rho)$. Taking a subsequence, if necessary, we can assume that if $n \geq N$ and $m \geq N$, then $\rho(u^{(m)}, u^{(n)}) < 1/2^{N+1}$. Let us chose irreducible representations $(y^{(m)}|x^{(m)})$ of $u^{(m)}$ such that

$$\|(y^{(m+1)}|x^{(m+1)}) - (y^{(m)}|x^{(m)})\| = \rho(u^{(m+1)}, u^{(m)}) < 1/2^{m+1}.$$

So if $n \geq N$ and $m \geq N$, then

$$\|(y^{(m)}|x^{(m)}) - (y^{(n)}|x^{(n)})\| < 1/2^N.$$

Hence, $(y^{(m)}|x^{(m)})$, $m \in \mathbb{N}$ is a Cauchy sequence in $\ell_1(\mathbb{Z})$ and so it has a limit point $z^{(0)} = (y^{(0)}|x^{(0)})$. Let $z_i^{(m)}$ be the $i$th coordinate of $z^{(m)} = (y^{(m)}|x^{(m)})$, $i \in \mathbb{Z}_0$, that is, $z_i^{(m)} = x_i^{(m)}$ if $i > 0$ and $z_i^{(m)} = y_{-i}^{(m)}$ if $i < 0$. Clearly that $z_i^{(m)} \to z_i^{(0)}$ as $m \to \infty$. We claim that if $z_i^{(0)} = c \neq 0$ then there is a number $N$ such that for every $m > N$, $z_i^{(m)} = c$. Indeed, it it is not so, then for every $n, m > N$, $\rho(u^{(m)}, u^{(n)}) > c$ and we have a contradiction.

For a given $\varepsilon > 0$ we denote by $z^\varepsilon$ a vector in $\ell_1(\mathbb{Z}_0)$ such that $z^\varepsilon$ has a finite support, $z_i^\varepsilon = z_i^{(0)}$ or $z_i^\varepsilon = 0$ and

$$\rho(z^\varepsilon, z^{(0)}) < \frac{\varepsilon}{3}.$$

Note that for this case $\rho(z^\varepsilon, z^{(0)}) = \|z^\varepsilon - z^{(0)}\|$. Let $N$ be a number such that for every $n > N$, $z_i^\varepsilon = z_i^{(n)}$ for all $i \in \text{supp}\, z^\varepsilon$ and $\|z^{(n)} - z^{(0)}\| < \frac{\varepsilon}{3}$. So

$$\rho(z^{(n)}, z^\varepsilon) = \|z^\varepsilon - z^{(n)}\| \leq \|z^\varepsilon - z^{(0)}\| + \|z^{(n)} - z^{(0)}\| < \frac{2}{3}\varepsilon.$$

Thus

$$\rho(z^{(n)}, z^{(0)}) \leq \rho(z^{(n)}, z^\varepsilon) + \rho(z^\varepsilon, z^{(0)}) < \varepsilon.$$

$\square$

### 2.2.3. Invertibility and Homomorphisms

If $u \in \mathcal{M}$ has an inverse with respect to the multiplication '$\diamond$' we denote it by $u^{-1} = u^{\diamond(-1)}$, that is,

$$u \diamond u^{-1} = u^{-1} \diamond u = \mathbb{I}.$$

**Proposition 9.** *Let $u \in \mathcal{M}$ and $\|u\| < 1$. Then $\mathbb{I} \bullet u^-$ is invertible in $\mathcal{M}$.*

**Proof.** It is easy to check that the proof for classical Banach algebras can be literally repeated for this case. In particular,

$$(\mathbb{I} \bullet u^-)^{-1} = \overset{\infty}{\underset{n=0}{\bullet}}\, u^{\diamond n},$$

where $u^{\diamond 0} = \mathbb{I}$, $u^{\diamond n} = \underbrace{u \diamond \cdots \diamond u}_{n}$ and the series on the right converges in $\mathcal{M}$. $\quad\square$

Next we consider ring homomorphisms and subrings of $\mathcal{M}$. In sequel we do not assume that ring homomorphisms preserve the multiplication by constants. Note that an element $x$ of a commutative Banach algebra $A$ is invertible if and only if $\varphi(x) \neq 0$ for every character $\varphi$ of $A$. The situation in $\mathcal{M}$ is different. Let $\mathbb{I}^{\bullet n} = \underbrace{\mathbb{I} \bullet \cdots \bullet \mathbb{I}}_{n} = (0 | \underbrace{1, \ldots, 1}_{n}, 0, \ldots)$.

**Proposition 10.** *Let $\varphi$ be a nonzero ring homomorphism from $\mathcal{M}$ to $\mathbb{C}$. Then $\varphi(\mathbb{I}^{\bullet n}) = n$ but $\mathbb{I}^{\bullet n}$ is non invertible for $n > 1$.*

**Proof.** Clearly, $\varphi(\mathbb{I}^{\bullet n}) = n\varphi(\mathbb{I}) = n$. On the other hand, $\mathbb{I}^{\bullet n} \diamond u = u^{\bullet n} \neq \mathbb{I}$ for every $u \in \mathcal{M}$. $\quad\square$

**Example 1.** *The following maps are ring homomorphisms from $\mathcal{M}$ to $\mathbb{C}$.*

1.  *Polynomials $T_n$, $n \in \mathbb{N}$ are (continuous) complex valued ring homomorphism of $\mathcal{M}$ but only $T_1$ preserves the multiplication by constants.*
2.  *Let $u = [(y|x)] \in \mathcal{M}$. We define*

$$\Theta(u) = \sum_{n=1}^{\infty} |x_n| - \sum_{n=1}^{\infty} |y_n|.$$

*Clearly, $\Theta$ is well defined. The additivity and multiplicativity will be proved for more general case.*

As usual $\mathcal{R}$ is a *subring* of $\mathcal{M}$ if it is a subset of $\mathcal{M}$ and a ring with respect to '$\bullet$' and '$\diamond$'. For example, let $\mathcal{M}_{00}$ consists of all elements $u = [(y|x)]$ such that if $(y|x)$ is irreducible, then supp $x$ and supp $y$ are finite sets. Then $\mathcal{M}_{00}$ is a dense subring of $\mathcal{M}$. We consider some nontrivial examples of closed subrings of $\mathcal{M}$.

**Example 2.** *Let $\mathcal{M}_\Delta$, $\mathcal{M}_S$ and $\mathcal{M}_1$ be defined by*

$$\mathcal{M}_\Delta = \{u \in \mathcal{M} : |x_j| \leq 1, \ |y_j| \leq 1 \ \forall \ \text{irreducible representations } (y|x) \in u\},$$

$$\mathcal{M}_S = \{u \in \mathcal{M} : |x_j| = 1 \text{ or } 0, \ |y_j| = 1 \text{ or } 0 \ \forall \ \text{irreducible representations } (y|x) \in u\} \cup \{0\},$$

$$\mathcal{M}_1 = \{u \in \mathcal{M} : x_j = 1 \text{ or } 0, \ y_j = 1 \text{ or } 0 \ \forall \ \text{irreducible representations } (y|x) \in u\} \cup \{0\}.$$

*Clearly, $\mathcal{M}_\Delta$, $\mathcal{M}_S$ and $\mathcal{M}_1$ are subrings of $\mathcal{M}$ and*

$$\mathcal{M}_\Delta \supset \mathcal{M}_S \supset \mathcal{M}_1.$$

*Also, $\mathcal{M}_1$ is isomorphic to the ring $\mathbb{Z}$ of integer numbers and the restriction of the topology of $(\mathcal{M}, \rho)$ to $\mathcal{M}_S$ and $\mathcal{M}_1$ coincides with the discrete topology. In the general case, let $U$ be a subset of $\mathbb{C}$. We denote by*

$$\mathcal{M}_U = \{u \in \mathcal{M} : x_j \in U, \ y_j \in U \ \forall \ \text{irreducible representations } (y|x) \in u\} \cup \{0\}.$$

*Then $\mathcal{M}_U$ is a subring of $\mathcal{M}$ if $U$ is closed with respect to the multiplication in $\mathbb{C}$ and $1 \in U$.*

**Proposition 11.** *Let $\gamma(t)$ be a function of one variable which is well defined and multiplicative on a subset $U \in \mathbb{C}$. We define*

$$\Theta_\gamma(u) = \sum_{n=1}^{\infty} \gamma(x_n) - \sum_{n=1}^{\infty} \gamma(y_n), \qquad u \in \mathcal{M},$$

*where* $(y|x) \in u$. *If U is closed with respect to the multiplication and* $1 \in U$, *then* $\Theta_\gamma$ *is a complex valued ring homomorphism of* $\mathcal{M}_U$.

**Proof.** Note first that $\Theta_\gamma(u)$ does not depend of the choice of a representative. Thus

$$\Theta_\gamma((y|x) \bullet (d|b)) = \Theta_\gamma(y \bullet d|x \bullet b)$$

$$= \sum_{n=1}^{\infty} \gamma(x_n) + \sum_{n=1}^{\infty} \gamma(b_n) - \sum_{n=1}^{\infty} \gamma(y_n) - \sum_{n=1}^{\infty} \gamma(d_n) = \Theta_\gamma(y|x) + \Theta_\gamma(d|b).$$

By the multiplicativity of $\gamma$ we have

$$\Theta_\gamma((0|x) \diamond (0|b)) = \sum_{n=i,j}^{\infty} \gamma(x_i)\gamma(b_j) = \sum_{n=i}^{\infty} \gamma(x_i) \sum_{n=j}^{\infty} \gamma(b_j)$$

and $\Theta_\gamma(x|0) = -\Theta_\gamma(0|x)$. So

$$\Theta_\gamma((y|x) \diamond (d|b)) = \Theta_\gamma((y \diamond b) \bullet (x \diamond d)|(x \diamond b) \bullet (y \diamond d))$$

$$= \sum_{n=i}^{\infty} \gamma(x_i) \sum_{n=j}^{\infty} \gamma(b_j) + \sum_{n=i}^{\infty} \gamma(y_i) \sum_{n=j}^{\infty} \gamma(d_j) - \sum_{n=i}^{\infty} \gamma(y_i) \sum_{n=j}^{\infty} \gamma(b_j) - \sum_{n=i}^{\infty} \gamma(x_i) \sum_{n=j}^{\infty} \gamma(d_j)$$

$$= \Theta_\gamma(y|x)\Theta_\gamma(d|b).$$

$\square$

**Example 3.** *Let us consider some examples of complex valued homomorphisms of subrings of* $\mathcal{M}$.

1.  *Let g be a multiplicative function from* $\mathbb{N} \to \mathbb{C}$. *In Number Theory such functions are called completely multiplicative arithmetic functions. Then for* $\gamma = |g|$, $\Theta_\gamma$ *is a complex valued ring homomorphisms of* $\mathcal{M}_S$ *and* $\mathcal{M}_1$.
2.  *Let* $\varepsilon < 1$ *and* $\varepsilon\Delta$ *be the closed disk in* $\mathbb{C}$ *of radius* $\varepsilon$, *centered at zero. Then* $\mathcal{M}_{\varepsilon\Delta}$ *is an ideal in* $\mathcal{M}_1$. *Let*

$$\chi_{\mathbb{C} \setminus \varepsilon\Delta}(t) = \begin{cases} 0 & \text{if } |t| \le \varepsilon \\ 1 & \text{if } |t| > \varepsilon, \end{cases}$$

*then* $\Theta_{\chi_{\mathbb{C} \setminus \varepsilon\Delta}}$ *is a complex valued ring homomorphisms of* $\mathcal{M}_1$. *Note that if* $u \in \mathcal{M}_1 \setminus \mathcal{M}_{\varepsilon\Delta}$ *and* $v \in \mathcal{M}_{\varepsilon\Delta}$, *then* $\rho(u,v) \ge \varepsilon$. *From here we have that* $\Theta_{\chi_{\mathbb{C} \setminus \varepsilon\Delta}}$ *is continuous.*

We do not know whether or not every complex valued homomorphism of $\mathcal{M}$ or its closed subring is continuous.

### 2.2.4. Additive Operator Calculus

Let $\Phi \colon \mathcal{M} \to \mathcal{M}$ be an additive map. Since it is a homomorphism of the additive group $(\mathcal{M}, \bullet)$ to itself, $\Phi$ is continuous at every point if and only if it is continuous at a point in $\mathcal{M}$. Let $\gamma \colon \mathbb{C} \to \mathbb{C}$ be an arbitrary function. Then it is well defined the following additive map from $\mathcal{M}_{00}$ to itself:

$$\Phi_\gamma(u) = (\dots, \gamma(y_n), \dots, \gamma(y_1)|\gamma(x_1), \dots, \gamma(x_n), \dots). \tag{13}$$

**Proposition 12.** *If there are constants $C > 0$ and $m \in \mathbb{N}$ such that $|\gamma(t)| \leq C|t|^m$, then $\Phi_\gamma$ is continuous, additive and well defined on $\mathcal{M}$.*

**Proof.** For every $u \in \mathcal{M}$

$$\|\Phi_\gamma(u)\| = \sum_{n=1}^\infty \|\gamma(x_n)\| + \sum_{n=1}^\infty \|\gamma(y_n)\| \leq C \sum_{n=1}^\infty (|x_n|^m + |y_n|^m) < \infty.$$

If $\|u\| < \varepsilon < 1$, then $\|\Phi_\gamma(u)\| < C\varepsilon$ and so $\Phi_\gamma$ is continuous at zero. Thus it is continuous. $\square$

**Example 4.** *(Power operators.) Let $m \in \mathbb{N}$. Then $\gamma(t) = t^m$ satisfies Proposition 12 and so the map $\Phi_m \colon u \mapsto u^m$, where $u = [(y|x)] \in \mathcal{M}$ and*
$$u^m = (\ldots, y_n^m, \ldots, y_1^m | x_1^m, \ldots, x_n^m, \ldots)$$

*is a continuous additive operator on $\mathcal{M}$.*

Let $k \in \mathbb{N}$ and
$$\sqrt[k]{a} = (a)^{1/k} = (a^{(1/k,1)}, a^{(1/k,2)}, \ldots, a^{(1/k,k)}), \qquad a \in \mathbb{C}$$

*be the multi-valued $k$th power root function. Let us consider*

$$(a^{(1/k,1)}, a^{(1/k,2)}, \ldots, a^{(1/k,k)}) = (a^{(1/k,1)}, a^{(1/k,2)}, \ldots, a^{(1/k,k)}, 0, 0 \ldots)$$

*as an element in $\ell_1$. Then, for every $u \in \mathcal{M}$ such that for an irreducible representation $(y|x)$ of $u$*

$$\sum_{n=1}^\infty (|x_n|^{1/k} + |y_n|^{1/k}) < \infty$$

*we can define*
$$\Phi_{1/k}(u) = u^{1/k} = (\cdots \bullet y_n^{1/k} \bullet \cdots \bullet y_1^{1/k} | x_1^{1/k} \bullet \cdots \bullet x_n^{1/k} \bullet \cdots).$$

The map $\Phi_{1/k}$ for $k > 1$ is a discontinuous additive operator, defined on a dense subset of $\mathcal{M}$. But if $m > k$, then we can define an additive operator
$$\Phi_{1/k} \circ \Phi_m$$

which is continuous on $\mathcal{M}$. Note that $\Phi_{1/k} \circ \Phi_k \neq \Phi_1$ if $k > 1$ because $\Phi_1$ is the identical operator while

$$\Phi_{1/k} \circ \Phi_k(u) = \underbrace{u \bullet \cdots \bullet u}_{k} = u \diamond \mathbb{I}^{\bullet k}.$$

We say that a map $A \colon \mathcal{M} \to \mathcal{M}$ is a *linear operator* if it is additive, preserves multiplications by constants, that is, $A(\lambda u) = \lambda A(u)$, $\lambda \in \mathbb{C}$ and if $A(u^-) = (A(u))^-$ for all $u \in \mathcal{M}$. From Proposition 12 it follows that there are a lot of additive operators. Linear operators, in contrast, can be described in a simple way.

**Theorem 7.** *Let $A$ be a continuous linear operator from $\mathcal{M}$ to itself. Then there exists an element $v \in \mathcal{M}$ such that*

$$A(u) = v \diamond u, \qquad u \in \mathcal{M}.$$

**Proof.** Let $u = \mathbb{I} = [(0|1,0,\ldots)]$ and $A(u) = [(b|a)] \in \mathcal{M}$. Set $v = [(b|a)]$. Let now $u$ be an element in $\mathcal{M}$ which can be represented by a vector $(y|x)$ with finite support

$$(y|x) = (\ldots, 0, y_m, \ldots, y_1 | x_1, \ldots, x_n, 0, \ldots).$$

Then we can write

$$[(y|x)] = y_1 \mathbb{I}^- \bullet \cdots \bullet y_m \mathbb{I}^- \bullet x_1 \mathbb{I} \bullet \cdots x_n \mathbb{I}$$

and so

$$A(u) = y_1 u^- \bullet \cdots \bullet y_m u^- \bullet x_1 u \bullet \cdots x_n u = v \diamond u.$$

Since the set $\mathcal{M}_{00}$ of elements with finite supports is dense in $\mathcal{M}$ and $A$ is continuous, $A = v \diamond u$ for every $u \in \mathcal{M}$. $\square$

We denote by $A_v(u)$ the operator $u \mapsto v \diamond u$, $u \in \mathcal{M}$. Let us prove some natural properties of operators $A_v$.

**Proposition 13.** *1. The operator $A_v$ is bijective if and only if $v$ is invertible in $\mathcal{M}$.*
*2. If the operator $A_v$ is surjective, then it is bijective.*
*3. The operator $A_v$ is injective if and only if $\ker A_v = 0$.*
*4. If $u \in \ker A_v$ for some $u \neq 0$, then $T_n(v) = 0$ for some $n \in \mathbb{N}$.*
*5. If $T_n(v) = 0$ for some $n \in \mathbb{N}$, then $A_v$ is not surjective.*

**Proof.** (1) If $v$ is invertible, then $A_{v^{-1}} = A_v^{-1}$ so $A_v$ is a bijection. Let now $B = A_v^{-1}$. Then from the Open Map theorem for complete metric groups (see [23]) it follows that $B$ is continuous. From Theorem 7 we have that $B = A_w$ for some $w \in \mathcal{M}$. Since

$$A_v \circ A_w = A_{v \diamond w} = A_{\mathbb{I}},$$

$w = v^{-1}$.

(2) Let $A_v$ be surjective. Then there exists $u \in \mathcal{M}$ such that $A_v(u) = v \diamond u = \mathbb{I}$. So $v$ is invertible and $A_u = A_v^{-1}$.

(3) If $A_v$ is injective, then $\ker A_v = 0$. Conversely, If there are $u, w \in \mathcal{M}$, $u \neq w$ such that $A_v(u) = A_v(w)$, then $A_v(u \bullet w^-) = 0$ and so $\ker A_v$ is nontrivial.

(4) If $u \in \ker A_v$, then $v \diamond u = 0$ and so

$$T_k(v \diamond u) = T_k(v) T_k(u) = 0, \qquad k \in \mathbb{N}.$$

Since $u \neq 0$, there exists a number $n \in \mathbb{N}$ such that $T_n(u) \neq 0$. So $T_n(v) = 0$.

(5) If $A_v$ is surjective, then it is bijective and so $v$ is invertible. But

$$1 = T_n(\mathbb{I}) = T_n(v \diamond v^{-1}) = T_n(v) T_n(v^{-1}) = 0,$$

a contradiction. $\square$

Note that for $v = \mathbb{I}^m$ the operator $A_v$ is not surjective but it is injective because

$$A_v(u) = \underbrace{u \bullet \cdots \bullet u}_{m}, \qquad u \in \mathcal{M}$$

and $T_k(v) = m > 0$ for every $k$. On the other hand for $v = (\ldots, 0, 1, 2|3, 0 \ldots)$, $T_1(v) = 0$ and so $A_v$ is not surjective but it is injective. Indeed, it is easy to check that $T_k(v) \neq 0$ for $k > 1$. So, if $v \diamond u = 0$ for

some $u = (y|x) \in \mathcal{M}$, then $F_k(x) = F_k(y)$ for $k > 1$. But from [13] it follows that also $F_1(x) = F_1(y)$ and so $T_k(u) = 0$ for all $k \in \mathbb{N}$, that is $u = 0$. Finally, for $v = (\ldots, 0, -1|1, 0 \ldots)$, $A_v$ has a nontrivial kernel which contains $u = (\ldots, 0|1, -1, 0 \ldots)$.

## 3. Discussion

According to Gelfand's theory, every commutative semi-simple algebra Fréchet $\mathcal{A}$ can be represented as an algebra of continuous functions on its spectrum $M(\mathcal{A})$ (see e.g., [24] p. 217, p. 231). If $\mathcal{A}$ consists of analytic functions on a Banach space $X$, then for every $x \in X$ the point evaluation functional $\delta_x$ belongs to $M(\mathcal{A})$. The map $x \mapsto \delta_x$ is one-to-one if and only if $\mathcal{A}$ separates points of $X$, for example, if $\mathcal{A} = H_b(X)$ is the algebra of all analytic functions of bounded type on $X$. Investigations of the spectrum of $H_b(X)$ were started by Aron, Cole and Gamelin in their fundamental work [2]. Note that, in the general case, $M_b = M(H_b(X))$ has complicated topological and algebraic structures (see [5,25]) which can be described only implicitly involving such tools as the Aron-Berner extension, topological tensor products, StoneČech compactification, ect. On the other hand, it is convenient for applications to have algebras of analytic functions of infinite many variables whose spectra admit explicit descriptions. If a subalgebra $\mathcal{A}$ of $H_b(X)$ has an algebraic basis of polynomials $P_1, P_2, \ldots, P_n, \ldots$, then every $\varphi \in M(\mathcal{A})$ is completely defined by its values on this basis, $\xi_1 = \varphi(P_1), \xi_2 = \varphi(P_2), \ldots, \xi_n = \varphi(P_n), \ldots$. So we can describe $M(\mathcal{A})$ as a subset of a sequence space $\{(\xi_1, \ldots, \xi_n, \ldots) : \xi_j \in \mathbb{C}\}$. Moreover, if $\|P_n\| = 1$ and $\deg P_n = n$, then it is not difficult to check that sequences $(\xi_n)$ should satisfy the following condition

$$\sup_n |\xi_n|^{1/n} < \infty. \tag{14}$$

Note that for the algebra of symmetric analytic functions of bounded type on $L_\infty[0, 1]$ condition (14) is sufficient [14] but for the algebra $H_{bs}(\ell_1)$ is not [12]. In the present paper we use this approach for $H_b^{sup}$ which is a subalgebra of $H_b(\ell_1(\mathbb{Z}_0))$ generated by polynomials $T_1, T_2, \ldots$. We can see that $H_b(\ell_1(\mathbb{Z}_0))$ is quite different than $H_{bs}(\ell_1)$. For example, the homomorphism defined by $T_k \mapsto -T_k$ is continuous in $H_b(\ell_1(\mathbb{Z}_0))$, while $F_k \mapsto -F_k$ is discontinuous in $H_{bs}(\ell_1)$. On the other hand, the homomorphism defined by $T_k \mapsto \lambda T_k$ is discontinuous for $\lambda \notin \mathbb{Z}$ and so the set of sequences $\xi_1 = \varphi(T_1), \xi_2 = \varphi(T_2), \ldots, \xi_n = \varphi(T_n), \ldots, \varphi \in M_b^{sup}$ does not support multiplications by constants. From here we have that condition (14) is not sufficient for description of $M_b^{sup}$.

The results of Section 2.2 show that the spectrum of $H_b^{sup}$ admits an interesting algebraic structure of commutative ring with respect to operations '$\bullet$' and '$\diamond$' which play roles of addition and multiplication. Using these operations and the $\ell_1$-norm we introduced a natural metric $\rho$ on $\mathcal{M}$, and proved that $(\mathcal{M}, \rho)$ is a complete metric space. We studied homomorphisms of $\mathcal{M}$ and described all linear operators of $\mathcal{M}$ to itself. So obtained results may be interesting in the theory of commutative topological algebras and for algebras of analytic functions on Banach spaces as well.

Supersymmetric polynomials and analytic functions are applicable in other branches of Mathematics and in Physics. Note first that supersymmetric polynomials of several variables were studied by many authors and in [18–20] we can find analogs of Formulas (7) and (8) for these cases (with using some different notations). Here we proved such results for infinite many variables and due to $\ell_1$-topology we can claim that $\mathcal{W}(y|x)(t)$ is a rational function, where the numerator and the denominator are functions of exponential type for every fixed $(y|x) \in \ell_1(\mathbb{Z}_0)$. But an important difference between finite- and infinite-dimensional case is that in the finite-dimensional case we can not to use the operations '$\bullet$' and '$\diamond$' because they do not preserve the dimension of the underlying space. Some applications of supersymmetric polynomials for Brauer groups are described in [26]. It seems to be that $H_b^{sup}$ can be applied for infinite generated Brauer groups in a similar way. Another application can be obtained for Statistical Mechanics.

In [27] we can find an approach to how classical symmetric polynomials can be used to modeling the behavior of ideal gas. According to this approach and using our notations, independent variables $x_1, x_2, \ldots$ correspond to abstract energy levels which particles of ideal gas may occupy; symmetric monomials

$$\sum_{i_1 < \cdots < i_n} x_{i_1}^{k_1} \cdots x_{i_n}^{k_n}$$

correspond to occupation these energy levels by particles; generating functions $\mathcal{G}(x)(t)$ and $\mathcal{H}(x)(t)$ correspond to grand canonical partition functions for bosons and fermions respectively, and Equation (2) is modeling the Bose-Fermi symmetry law. From this point of view and taking into account (7), supersymmetric polynomials may be useful for the description of ideal gas consisting of both type particles: bosons, and fermions. Moreover, the Bose-Fermi symmetry in our notations means just $[(x|x)] = 0$.

Note that Statistical Mechanics work with the situation when the number of particles, $N$ tends to infinity. The fact that we consider the closure of polynomials in a metrizable topology allows us to proceed with limit values as $N \to \infty$. The $\ell_1$-topology of the underlying space $\ell_1(\mathbb{Z}_0)$ is guarantying that all abstract supersymmetric polynomials are well defined on this space. For example, if we will use $\ell_2(\mathbb{Z}_0)$ instead of $\ell_1(\mathbb{Z}_0)$, then $T_1$ will be not defined. Finally, we can expect that the algebraic operations '•' and '⋄' may have a physical meaning in the proposed approach. But such kind of problems is outside of the topic of our article.

## 4. Conclusions

In this article, we considered the algebra $H_b^{sup}$ of analytic functions of bounded type generated by supersymmetric polynomials on $\ell_1(\mathbb{Z}_0)$. We have described some algebraic bases of the subalgebra of supersymmetric polynomials and corresponding generating functions. Such a description is important in order to study the spectrum (the set of complex homomorphisms) of $H_b^{sup}$. In particular, it is shown that every point evaluation complex homomorphism can be represented as a ratio of two entire functions of exponential type. Also, we constructed an example of complex homomorphism which is not a point evaluation functional. However, we have not a complete description of the spectrum of $H_b^{sup}$. In particular, it is unclear under which conditions a meromorphic function is of the form (12) for some $\varphi \in M_b^{sup}$? Note that such kind of problem is also open for the algebra $H_{bs}(\ell_1)$ [10,12].

Our goal is establishing the structure of a complete metric commutative ring on the set $\mathcal{M}$ of point evaluation functionals of $H_b^{sup}$. The algebraic structure of $\mathcal{M}$ is very close to the Banach algebra structure but $\mathcal{M}$ is not a Banach algebra because it is not a linear space. So we have a natural question: which Banach algebras properties can be extended to the ring $\mathcal{M}$? For example, we can see that if an element is closed to the unity, then it is invertible. But we do not know: do $\mathcal{M}$ admits a discontinuous complex homomorphisms? Also, we investigated homomorphisms of $\mathcal{M}$, its subrings and additive operators of $\mathcal{M}$. The role of obtained results in the theory of algebras of analytic functions on Banach spaces and possible applications in Physics are discussed.

**Author Contributions:** These authors contributed equally to this work.

**Funding:** The second author was partially supported by Ministry of Education and Science of Ukraine Grant 0119U100063.

**Conflicts of Interest:** The authors declare no conflict of interest.

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
