# Peer review of "Supersymmetric Polynomials on the Space of Absolutely Convergent Series"

_symmetry, doi:10.3390/sym11091111_

Round 1
Reviewer 1 Report
Referring to the Manuscript ID: symmetry-558423 review report: 1-The article structure is quite appropriate and good organized, with minor lacks. The introductory part made a tour of works in this domain: Banach spaces, homogenous polynomials, supersymmetric polynomials. Entire work is a course of advanced math, especially algebraic domain. 2-The article is written for high-level mathematicians. Even there are many examples and proofs it's obvious that a section named Results and Discussion and other one named Conclusions are necessary, as a proposal, not as a request. The authors start to cite works from References with # [10] and continue with [2,3,18,22]. There is no rule to start with [1], but usually, the authors respect this.Author Response
Response to Reviewer 1 Comments
Point 1
Section named Results and Discussion and other one named Conclusions are necessary, as a proposal, not as a request.
Response 1: Yes, in the revised version we included sections Results, Discussion, and Conclusions.
Point 2
The authors start to cite works from References with # [10] and continue with [2,3,18,22]. There is no rule to start with [1], but usually, the authors respect this.
Response 2: We used the alphabetic order for References, but according to the journal's rules and recommendations of the Referee, in the revised version we use the order of appearance in the text.
Reviewer 2 Report
The work is about an algebra of analytic functions on the Banach space, which is investigated by using infinite dimensional complex analysis and the theory of Frèchet. The metric of algebra spectrum about supersymmetric analytic functions is analyzed.
The first part of the work is devoted to previous existing results, they could be, in any cases, omitted, please revise,
Many theoretical results, in particular about proofs haven't so important theoretical basis, so they could fall in trivial results (see the Section 2, i.e.), this can make the work poreer than it could be, if one consider the big amount of the shown results, please re-consider the unnecessary proofs..
Neverthenless I consider the results very complete by looking at the whole plan of the work.
The works takes into account an important generality of methods, if one see Proposition 6 for instance, it considers some behaviours very related to the algebras of the considerd functions.
I consider important also the kind of topology taken into account, in fact that generated by ρ is absolutely lighting for the topic.
I very appreciate the provided examples, they are representative of the abstactive argument.
References are complete and well choosen.
Minor remarks:
Please check the definition of symbols all over the paper.
A "conclusions and future developments" section could be added.
Author Response
Response to Reviewer 2 Comments
Point 1
Many theoretical results, in particular about proofs haven't so important theoretical basis, so they could fall in trivial results (see the Section 2, i.e.), this can make the work poreer than it could be, if one consider the big amount of the shown results, please re-consider the unnecessary proofs..
Response 1: Not complicated results as Theorem 1, Propositions 1, 3, 6, Corollaries 1,4 have no proofs or just sketch of proofs. On the other hand, complete proofs of more complicated statements are convenient for readers and since there is no restriction for the size of papers, we try to give the proofs.
Point 2
Please check the definition of symbols all over the paper.
Response 2: Yes, we checked it and corrected where it was necessary.
Point 3
A "conclusions and future developments" section could be added.
Response 3: Yes, in the revised version we included sections Results, Discussion, and Conclusions.
Reviewer 3 Report
The paper deals with a subalgebra of entire functions of bounded type that is generated by supersymmetric polynomials. The authors quite accurately and consistently present proofs of their results, but I have doubts about the relevance of this work and contribution to science required by any published scientific paper. I think that the authors should give important examples proving the value of their results for theory and applications. Of course, such examples should be preceded by a serious discussion of the motivation for conducting this study.
Only after making major corrections will I be able to continue the review process.
Author Response
Response to Reviewer 3 Comments
Point 1
I think that the authors should give important examples proving the value of their results for theory and applications. Of course, such examples should be preceded by a serious discussion of the motivation for conducting this study.
Response 1: According to Journal Rules “Symmetry is … covering research on symmetry phenomena wherever they occur in mathematical and scientific studies.” The main object of the presented paper is the symmetry phenomena in algebras of polynomials and analytic functions on infinite-dimensional Banach spaces. The considered problems (like descriptions of algebraic and topological structures of spectra) are motivated by the inner logic of development of the theory. However, we understand that “Symmetry” is a multidisciplinary journal with a large auditory and so the article should be interesting not only for pure mathematicians. In the revised version we included sections Results, Discussion, and Conclusions. In section Discussion there is more detailed information about related questions in the theory of topological algebras and algebras of analytic functions on Banach spaces, and about the role of our results in this context. Also, we considered some possible applications in Statistical Physics. We do not know: what is the physical meaning of elements of algebraic or topological structures of our objects? Actually, it depends on concrete physical interpretation. But we propose a tool, which can be useful for this purpose.
In Section Conclusion, there are some concluding remarks and open questions for further development.
Round 2
Reviewer 3 Report
After making corrections, the paper has become better and can be recommended for publication.
However, in some places more explanations or details should be given for better readability.
Comments:
(1) The citation to [24] (Page 15): Since [24] is a rather thick book, it would be more reader-friendly to provide concrete Section or page numbers.
(2) Page 4. In the proof of Proposition 1, the corresponding references should be given.
(3) Page 4. It is recommended to change "Items (1)-(3) are straightforward results of definitions" to "Assertions (1)--(3) are straightforward consequences of definitions".
(4) Page 5. The phrase "From (10) we have (8)". Give more details.
Author Response
Response to Reviewer 3 Comments (Round 2)
Point 1
The citation to [24] (Page 15): Since [24] is a rather thick book, it would be more reader-friendly to provide concrete Section or page numbers.
Response 1: Yes, we added the page numbers
Point 2
Page 4. In the proof of Proposition 1, the corresponding references should be given.
Response 2: Yes, required references added
Point 3
Page 4. It is recommended to change "Items (1)-(3) are straightforward results of definitions" to "Assertions (1)--(3) are straightforward consequences of definitions".
Response 3: Done
Point 4
Page 5. The phrase "From (10) we have (8)". Give more details.
Response 4: Required details are given